# Analysis of the Clinical Status and Treatment of Facial Cellulitis of Odontogenic Origin in Pediatric Patients

**DOI:** 10.3390/ijerph20064874

**Published:** 2023-03-10

**Authors:** Adrianna Słotwińska-Pawlaczyk, Bogusława Orzechowska-Wylęgała, Katarzyna Latusek, Anna Maria Roszkowska

**Affiliations:** 1Department of Pediatric Otolaryngology, Head and Neck Surgery, Chairs of Pediatric Surgery, Medical University of Silesia, 40-055 Katowice, Poland; 2Ophthalmology Clinic, Department of Biomedical Sciences, University Hospital of Messina, 98122 Messina, Italy

**Keywords:** oral hygiene, dental caries, odontogenic infection, children, pediatric patients, facial cellulitis

## Abstract

The most common cause of the development of odontogenic infection is untreated dental caries, which initially leads to pulpitis. If an odontogenic infection is left untreated, it will pass through the limiting bone plate and will infiltrate deeper structures. Odontogenic infections are different in adults and children. The study was conducted at the Department of Pediatric Otolaryngology and Pediatric Head and Neck Surgery of Upper Silesian Children’s Health Center in Katowice in the 2020–2022. We included 27 patients aged 2–16 in the study. Patients were diagnosed with an active, acute odontogenic inflammatory process in the head and neck area. We assessed pain, trismus, extraoral and intraoral swelling and the level of CRP [C Reactive Protein], WBC [White Blood Cells], NLR [Neutrophil Lymphocyte Ratio], D-dimers and Prealbumins. The results were analyzed in terms of the location of the source of inflammation: maxilla or mandible and the type of source of infection: deciduous tooth or permanent tooth. Deciduous teeth are more often the cause of odontogenic infection in the maxilla, while permanent teeth in the mandible. Trismus, extraoral, and intraoral swelling occurred in all infections caused by permanent teeth. The CRP and NLR ratio is statistically higher in infection, which originates from permanent teeth. The mean hospitalization time was also longer for infections from permanent teeth 3.42 days than for deciduous teeth 2.2 days. The varied clinical picture of odontogenic infections in children requires periodic analyzes of statistical data related to epidemiology, etiology, and symptomatology in order to update diagnostic and therapeutic procedures.

## 1. Introduction

Caries is an infectious disease and therefore can spread rapidly and extensively in teeth [1]. In addition, it is considered a civilization disease among all diseases of the oral cavity. According to the Polish Central Statistical Office, the number of children with caries aged 3–6 years in 2012 was almost 1.7 million, which is approximately 4.4% of the entire population [2]. Poland is not the only country struggling with this problem, as it also affects highly developed countries such as the United States, Great Britain, France, and Germany [3]. Untreated caries of deciduous and permanent teeth can lead to premature loss of teeth, as well as disorders of the whole organism [4]. The most common cause of the development of odontogenic infection is untreated dental caries, which initially leads to pulpitis. Other causes include necrosis of the dental pulp as a result of trauma, periodontitis, or inflammation of the peri coronal tissues associated with difficult tooth eruption [5]. If an odontogenic infection is left untreated, it will pass through the limiting bone plate and infiltrate deeper structures, passing through superficial anatomical spaces, reaching deep spaces [6]. In addition, odontogenic infections can spread through the blood and lymphatic vessels [7]. The incidence of deep head and neck infections has decreased in recent years due to the use of antibiotic therapy. However, we should still remember about possible serious complications: facial cellulitis, cavernous sinus thrombosis, airway obstruction, Ludwig’s angina, soft tissue abscesses, necrotizing fasciitis, meningitis, brain abscesses, mediastinitis or sepsis [8]. Odontogenic cellulitis accounted for 50% of all facial infections in the study by Biederman et al. [9] and almost 54% of pediatric patients with facial odontogenic cellulitis required hospitalization in the study by Kuo et al. [10]. Odontogenic infections are different in adults and children. The course of facial infections in children is more dynamic and acute, which is associated with easy spreading into deep spaces and leads more often to the appearance of systemic symptoms such as fever, dehydration, and impaired respiratory function [11]. Another difference is in the bacterial spectrum. In the group of adults, odontogenic soft tissue inflammation of the head and neck is multimicrobial with a significant component of anaerobic bacteria, while in children the most common causes are staphylococci and streptococci. Children are also more prone to the spread of infection through the lymphatic vessels, due to the higher content of this tissue in children compared to adults [12]. Treatment of odontogenic infections in children often requires multi-specialized treatment. The varied clinical picture and response to treatment make it difficult to develop specific algorithms for diagnostics and conservative and surgical treatment [13]. Most articles concern odontogenic infections in adults, while articles based on the pediatric population describe both odontogenic and non-odontogenic infections, which gives an unreliable clinical picture [14]. Additional studies and analyzes are necessary to update clinical, etiological, and epidemiological data on odontogenic infections in pediatric patients.

The aim of this investigation is to analyze the amount of odontogenic infections in 3 years in the Upper Silesian Children’s Health Center in Katowice, obtaining the epidemiological characteristics and tracing the path of the patients to calculate the percentage of serious complications and the mortality rate of acute cellulitis of the head and neck area.

## 2. Materials and Methods

Hospitalized patients with diagnosed odontogenic infections in the oral cavity were qualified for the study. The study was conducted at the Department of Pediatric Otolaryngology and Pediatric Head and Neck Surgery of Upper Silesian Children’s Health Center in Katowice in the 2020–2022. We included 27 patients aged 2–16 in the study. Qualified patients were diagnosed with an active, acute odontogenic inflammatory process in the head and neck area: teeth with gangrene of the pulp, chronic inflammation of the periapical tissues causing the appearance of abscesses in the head and neck area. The exclusion criteria of patients from the study were: systemic diseases, patients with non-odontogenic infections, rhinosinusitis, inflammation of the salivary glands, immunocompromised patients, patients after radiotherapy of the head and neck area, patients after systemic antibiotic therapy in the last 6 weeks, patients during steroid therapy and lack of consent of the legal guardian for the child’s participation in the study. The patients underwent an interview (A), physical examination (B), laboratory examinations (C), and imaging diagnostics (D) in order to assess odontogenic foci in the oral cavity.

(A): The VAS scale was used in children over 9 years of age, but in younger children, we relied on an interview with parents and their observation of the child at home.

(B): The presence of extraoral swelling was assessed on the first day of hospitalization and then as part of outpatient control approximately 7 days after the end of hospitalization. The largest dimension among the three examined dimensions was assessed: ear tragus-nasal wing; ear tragus-mouth angle; ear tragus-mental process. The difference between the second and first measurement was recorded. The second measurement was considered normal and showed no signs of oedema. If the difference was 0, we noted a lack of swelling. The study also evaluated the presence or absence of intraoral swelling. The number and type of occupied spaces was subjectively determined based on extraoral and intraoral oedema. The distance between the incisal edges of the lower and upper central incisors was measured to assess trismus.

(C): We assumed no trismus when the mouth opening was greater than 30 mm. The patients qualified for the study had 3 mL of peripheral blood taken to determine the level of CRP [C Reactive Protein], WBC [White Blood Cells], NLR [Neutrophil Lymphocyte Ratio], D-dimers, and Prealbumins. The post-extraction socket was collected for the assessment of the bacteriological spectrum by mass spectrometry. The results were analyzed in terms of the location of the source of inflammation (I): maxilla or mandible and the type of source of infection (II): deciduous tooth or permanent tooth. The time of hospitalization was also assessed.

This study was approved by the Ethics Committee of Medical University of Silesia (Protocol No. KNW/0022/KB1/76/19) and conducted in accordance with the World Medical Association Declaration of Helsinki.

The data was analyzed using Statsmodels (Python package) version 0.13.2. The Student’s *t*-test was used to compare dependent variables between the specified groups. The *p*-value < 0.05 was considered statistically significant.

## 3. Results

(IA): Odontogenic infections of the upper part of the face were reported in 48% of pediatric patients, who visited the Department, while in 52% of the lower part of the face. In the whole group of qualified patients, 52% were boys and 48% were girls. The mean age of the entire group was 8.56 ± 5.43. The mean value of the VAS scale for all patients over 9 years of age was 5.92 ± 1.07. A difference was shown in the subjective assessment of pain according to VAS in patients over 9 years of age, where the mean value for upper face infections was 7.00 and 5.6 for lower face infections.

(IB): The comparison of upper and lower facial infections for the study variables is presented in Table 1. The anterior and posterior deciduous teeth were the cause of odontogenic infections in the upper part of the face in 38.5% each, while in 23% the cause was the permanent anterior teeth (Figure 1). There was no infection from the upper posterior permanent tooth. The cause of odontogenic infection of the lower part of the face was only the posterior deciduous teeth in 36% and the posterior permanent teeth in 64% (Figure 2). The distribution of general symptoms in upper and lower facial infections is shown in Figure 3. Trismus was more common in lower face infections in 71% as well extraoral swelling was more common in lower face infections in 79% of cases. Gingival fistula and fever were more common in the maxilla, 46% and 31%, respectively. Difficulty swallowing was reported in 43% of patients with lower facial infection.

(IC): The analyzed biochemical parameters of CRP, NLR, WBC, and D-Dimer in the group of patients with infection of the upper and lower part of the face were above the norm or at the upper limit of normal values (WBC). Prealbumin levels were below normal, but there was no significant difference between the lower and upper parts of the face. In the case of odontogenic infection of the maxilla and mandible, the mean hospitalization time was 3.54 and 2.00 days, respectively, with a significance level of 0.08.

(IIA): A comparison of odontogenic infections with deciduous and permanent teeth as the source of infection is presented in Table 2. More often deciduous teeth in the maxilla were the cause of odontogenic infections in 73% of cases, while permanent teeth more often in the mandible in 75%.

(IIB): The distribution of general symptoms in odontogenic infections originated from deciduous and permanent teeth is shown in Figure 4. We present patients with facial cellulitis (Figure 5, Figure 6 and Figure 7). Trismus, extraoral and intraoral swelling occurred in all infections caused by permanent teeth. Gingival fistula and fever were also more common in permanent tooth infections (42%). Dysphagia occurred only in inflammations from permanent teeth and this symptom was not found in the case of infection from deciduous teeth.

(IIC): CRP, NLR, WBC, and D-Dimers were above the upper limit of normal values and prealbumin below the lower limit, which is similar to the previous comparison of biochemical parameters in Table 1. The NLR and CRP value is statistically higher during inflammation, which originates from permanent teeth. The mean hospitalization time was also longer for infections from permanent teeth 3.42 days and 2.2 days for deciduous teeth at a significance level of 0.18. 

We introduced a linear regression model to investigate the relationship between the length of hospital stay and the predictor variables. The NLR ratio was a significant predictor of hospital stay in upper-face odontogenic infections (*p* < 0.01). The coefficient of determination R^2 for this model was 0.5137, which means 51.37% of the variation in the length of stay could be assigned to the NLR value at admission (Figure 8). There was a linear relationship between CRP with a time of hospital stay in upper face odontogenic infections, i.e., the higher the CRP level, the longer hospitalization. This linear relationship means that 48.12% of the variation during hospital stay was explained by the CRP (Figure 9). In 35.17% the length of hospitalization in lower odontogenic infections could be predicted based on the NLR level at hospital admission (Figure 10), and only in 14.81% used the CRP level (Figure 11).

Treatment of acute odontogenic cellulitis consisted of removing the cause and symptomatic treatment. Causal teeth were removed in all patients and an incision and drainage of pus was performed in 19%. All permanent teeth including anterior teeth were qualified for removal because endo treatment had failed or the patient’s condition was serious (septic shock) and the cause of the infection had to be removed immediately. These procedures were performed in 19% of patients under local anesthesia, while 81% of patients required treatment under general anesthesia. Antibiotics were introduced in 78% of patients. All of these patients received intravenous antibiotics. Amoxicillin with clavulanic acid was used in 41% of the cases and amoxicillin with clavulanic acid with metronidazole in 19%, with clindamycin in 11% of patients (Figure 12). Antibiotic therapy was started after admission if it was indicated. Some patients were not qualified for antibiotic therapy because deciduous teeth were the source of the infection and the clinical condition was not very advanced.

Swabs were taken from post-extraction tooth socket for bacteriological analysis. Forty-seven bacterial strains were detected, including 29 G+ bacteria and 18 strains of G- bacteria. Twelve strains of Streptococcus and three strains of Staphylococcus which belong to facultative anaerobes were detected. Staphylococcus aureus was isolated most frequently (8.5%), followed by Actinomyces naeslundi (6.4%), Streptococcus anginosus (6.4%), Streptococcus mitis (6.4%) and Staphylococcus epidermidis (6.4%) (Table 3). Both facultative anaerobic G- and aerobic G- bacteria were detected. The most common aerobic G- bacterium was Neisseria subflava (10.6%), and the anaerobic bacterium was Haemophilus parainfluenzae (6.4%). Escherichia coli, Eikenella corrodens and Klebsiella pneumoniae were also found among G- anaerobic bacteria (Table 4).

## 4. Discussion

Odontogenic cellulitis is a deep inflammation of the subcutaneous tissue of an acute and diffuse nature, which spreads to other anatomical spaces as a result of the development of dental inflammation [6].

The alveolar bone is initially the largest barrier that inhibits the spread of inflammation. Subsequently, the inflammation spreads to the periosteum, which is thicker within the mandible. The inflammatory process that passes through the periosteum affects soft tissue, aponeurosis, and muscles [15]. Severely advanced odontogenic infections are most common in emergency departments and children’s maxillofacial surgery departments. A child with facial swelling and fever should undergo a thorough clinical examination. Failure to implement prompt and appropriate treatment may lead to serious complications: dehydration, central nervous system disorders, airway obstruction, or sepsis. In the study, we have not recorded patients with other chronic diseases as well as Doll et al. in their study [16].

Women slightly predominated in the study conducted by Lim et al. [17] as in our study (60%), while Kara et al. calculated the ratio of men to women as 1.4:1 [18]. Mean age values vary in many studies. Doll et al. reported an average age of 6.3 years [16], Lim et al. estimated at 6.36 years [17], Kuo et al. calculated a mean age of 5.17 years [10], while Thikkurissy et al. estimated at 8.3 years [19] and in our study it was 8.85 years. We recorded more patients with infection of the lower face (52%) than the upper face (48%). Al-Malik et al. said that the cause of infection was most often the posterior deciduous teeth [11] and more often the posterior teeth of the mandible than the maxilla were the causative teeth [11,20], which is consistent with the results of our study in relation to permanent teeth. Deciduous teeth more often than permanent teeth were the cause of the development of facial cellulitis (55%) as in the study by Ritwik et al. [21]. Lim et al. reported 62.1% positive pain results in patients qualified in the study [17], while in our study all patients reported pain. The VAS scale was used in children over 9 years of age and in younger children we relied on an interview with parents. According to Lin et al. [22] fever and facial swelling were the main factors indicating the need for hospital care. Swelling occurred in 76.8% of their patients, whereas in our study 70% of patients had extraoral oedema and 90% had intraoral oedema. Lim et al. observed extraoral swelling in 32% of patients and they diagnosed purulent discharge on admission to the hospital in 67% of children [17].

We found an active purulent fistula in 30% of cases. Lin et al. observed fever in 14.3% of patients in the 2006 study and 37.3% in 2013 [10,22], what is similar to our results (26%). Blood biochemical parameters such as CRP, NLR and WBC are used to assess the general and local condition of the patient. In this study, the biochemical inflammatory parameters were elevated and the mean values were: CRP 54.32 mg/L, WBC 10.93 × 10^3^/uL, and NLR 6.33. Kuo et al. assessed an average CRP of 27.49 mg/L and a WBC of 12.39 × 10^3^/mm^3^ [10]. Additionally, Donn et al. in their study confirmed the presence of elevated CRP and WBC in all types of odontogenic abscesses [16]. In a previous analysis we assessed a statistically significant positive relationship between CRP and extraoral swelling as well as the NLR correlates significantly with extraoral swelling and the length of hospitalization [23]

The aim of a therapy In hospitalized patients is elimination the cause of infection by removal a causal tooth and drainage by incision of the abscess if it is necessary. According to other authors, the faster the elimination of the source of infection, short the hospitalization time [16,22]. We found a difference in hospitalization time for odontogenic infections caused by a causal deciduous tooth and a permanent tooth, which was 2.2 days and 3.42 days, respectively (*p* = 0.18). Doll et al. showed a statistically significant relationship between the length of stay in the hospital and different age groups (*p* < 0.001), so that the length of hospitalization at the age of 14–17 years was twice longer than in patients younger than 6 years [16]. The mean length of hospital stay in the study by Ritwik et al. was 2.5 days [21], but Al-Malik et al. estimated the length of hospital stay more than 4 days [11], while Kuo et al. determined mean hospitalization time to 5.15 days [10]. He also assessed the mean length of hospital stay for patients with upper face infection as 5.10 days and lower face infection as 5.21 days at a significance level of *p* = 0.662. In our study, the average hospitalization was 3.54 days and 2.00 days, respectively (*p* = 0.08), so the difference in hospitalization time is greater than in Kuo et al. research.

The principles of treatment in odontogenic infections Ide tooth extraction or root canal treatment and drainage [24,25]. In addition, analgesics and antibiotic therapy are used in accordance with current guidelines as well as rehydration and nutrition [5]. Patients self-reported or were referred by a doctor of another specialty to our Department of Maxillofacial Surgery, where the therapeutic model is based on the removal of the odontogenic cause with possible additional drainage of pus. We removed the causal teeth in all patients and in 20% we additionally incised and drained the abscess. No patient was qualified for endodontic treatment due to advanced or severe infection or lack of parental consent. Causal teeth were extracted in 78.7% of cases in the study of Al-Malik et al. [11]. Lim et al. in 46.1% of cases used tooth opening for endodontic treatment as a method of treatment. In their study extraction or incision and drainage were performed under general anesthesia in 7.8% of cases and in 26.6% under local anesthesia [17]. In our study 75% of patients were treated under general anesthesia and 25% under local anesthesia.

The cause of odontoIenic infections are most often endogenous oral bacteria, but not introduced non-resident bacteria [11,26]. The most common bacterial agents associated with odontogenic infection are Streptococcus mutans (24.5%) and Porphyromonas gingivalis (23.6%). The most significant causative factors are Streptococcus salivarius (10.1%) and Streptococcus sanguis (8.2%) [6,27]. Broad-spectrum antibiotics are recommended for the treatment of odontogenic cellulitis in children. Amoxicillin or amoxicillin with clavulanic acid per os is recommended for a maximum of 5 days due to the antibiotic effect [28]. Lim et al. found that 24% of their patients took a drug prescribed by another doctor and in 86% of cases it was an antibiotic [17], which may be related to the severity of the disease and the indication for antibiotic therapy or overuse of antibiotic therapy. Complementary antibiotics are not corrected in immunocompetent patients after successful drainage [29]. The indication for the use of antibiotic therapy are symptoms of spreading infection such as fever or extraoral swelling. Another indication for antibiotic therapy is medium or severe immunocompetence of the patient [24,25]. Ritwick et al. determined two factors that were significantly correlated with shorter treatment of odontogenic infections in children: intravenous administration of antibiotics and deciduous teeth as an etiological factor [20]. The use of penicillin has a high success rate. In this study, 41% of the patients were treated with amoxicillin with clavulanic acid, 19% with amoxicillin with clavulanic acid with metronidazole and 11% with clindamycin. Cefazolin and vancomycin were used, when we obtained a positive result of the bacteriological examination. 22% of the patients did not require antibiotic therapy. According to the literature, amoxicillin is the antibiotic of choice and administered with metronidazole when we suspect an anaerobic component [24,25].

It is recommended to use clindamycin, azithromycin or moxifloxacin or targeted antibiotic therapy for odontogenic infections in the case of penicillin allergy [30,31]. The use of antibiotic therapy cannot be the only form of treatment, but only a therapy that supports the causal treatment.

A limitation of the present study is that the number of patients is not very large. The severity of the infection and the length of hospital stay are influenced by many complex factors. We evaluated several parameters, and therefore, further studies with larger groups of patients.

## 5. Conclusions

Treatment of odontogenic infections in children requires an interdisciplinary approach. The varied clinical pictures and responses to treatment are a challenge for general practitioners, dentists, and maxillofacial surgeons. We found no significant differences in upper and lower face infections. On the day of admission, precisely measured NLR and CRP levels in upper face odontogenic infections can be predictive factors for the length of hospitalization in patients hospitalized. Trismus and dysphagia are symptoms more common in infections of the lower face, while the fistula is more likely to occur in lower face odontogenic infections. Deciduous teeth were more often the cause of infection, which could be related to poor oral hygiene in younger children or an improper diet that leads to the appearance of untreated bottle caries. A statistically significant difference in CRP and NLR levels was observed between infections which source was a deciduous tooth and a permanent tooth. There was a difference in the time of hospitalization between patients with a deciduous teeth and a permanent teeth as the source of infection as well between patients with infections originated from maxilla and mandible. In the group of qualified pediatric patients, the most commonly isolated G+ bacteria was Staphylococcus aureus and G− bacteria was Neisseria subflava. The basis for therapeutic success is the correct diagnosis and quick implementation of causal therapy, especially in the population of pediatric patients. Antibiotic therapy should be used only in specific cases according to guidelines to reduce the risk of the appearance of antibiotic-resistant bacterial strains

## Figures and Tables

**Figure 1 ijerph-20-04874-f001:**
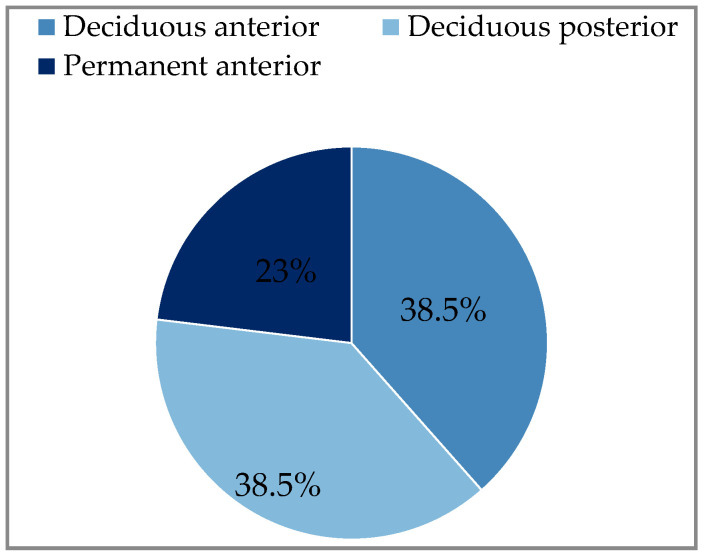
Upper face-source of infection (n = 13).

**Figure 2 ijerph-20-04874-f002:**
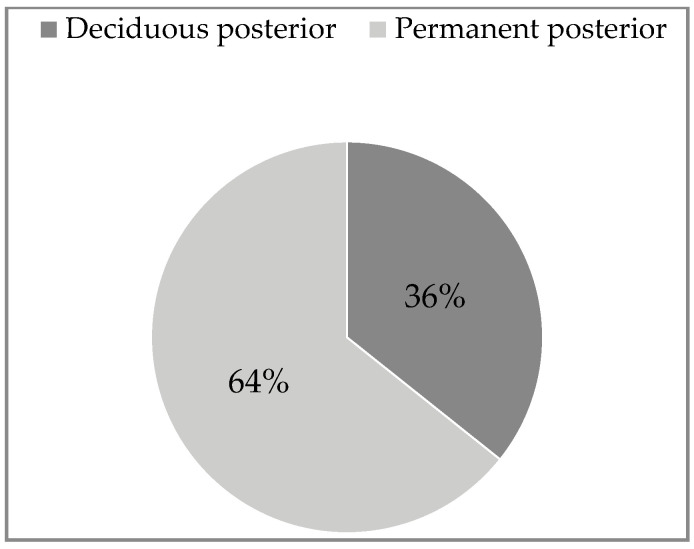
Lower face-source of infection (n = 14).

**Figure 3 ijerph-20-04874-f003:**
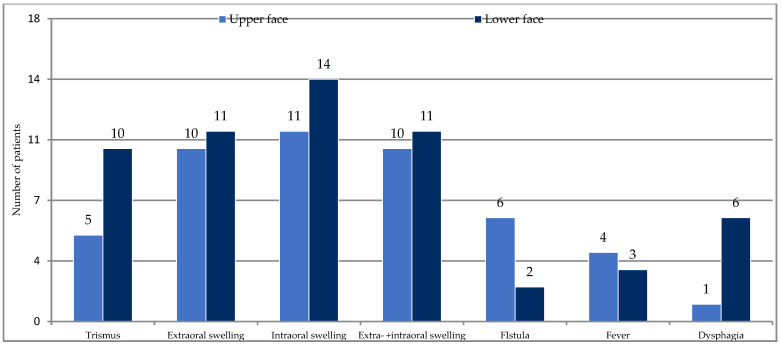
General symptoms in odontogenic infections of the upper and lower face.

**Figure 4 ijerph-20-04874-f004:**
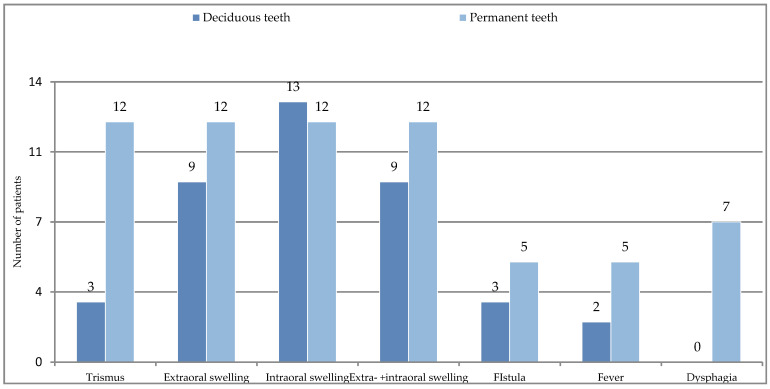
General symptoms in odontogenic infections of deciduous and permanent teeth.

**Figure 5 ijerph-20-04874-f005:**
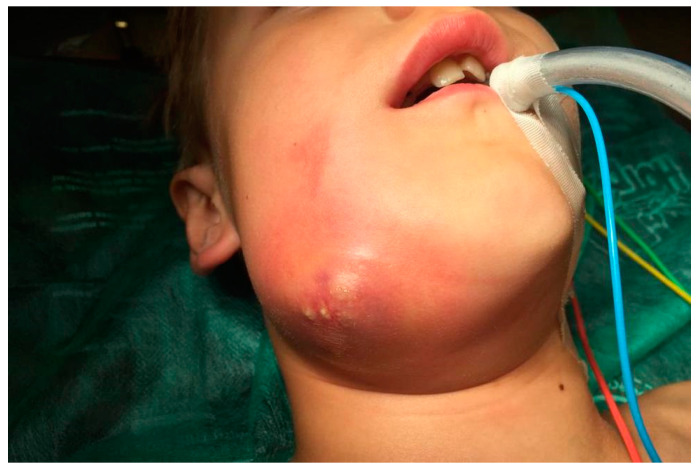
8-year-old patient with facial phlegmon on the right side.

**Figure 6 ijerph-20-04874-f006:**
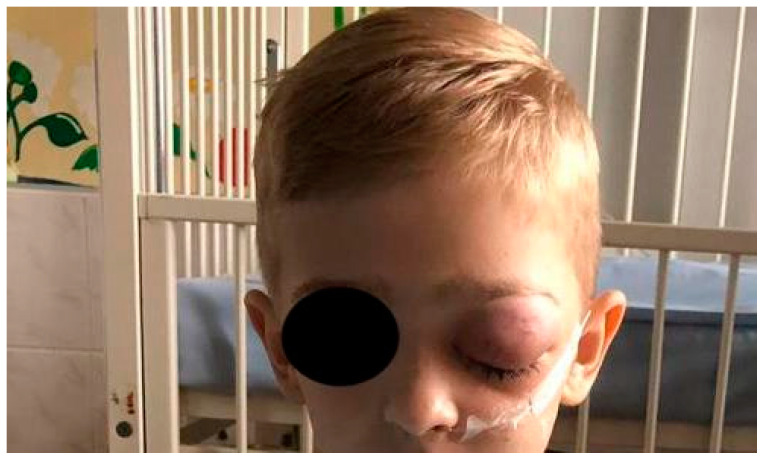
7-year-old patient with orbital phlegmon on the left side.

**Figure 7 ijerph-20-04874-f007:**
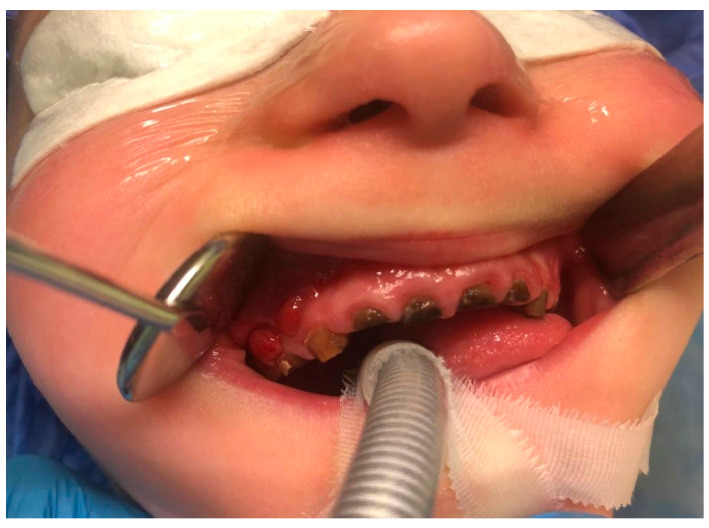
Odontogenic source of infection of the maxilla.

**Figure 8 ijerph-20-04874-f008:**
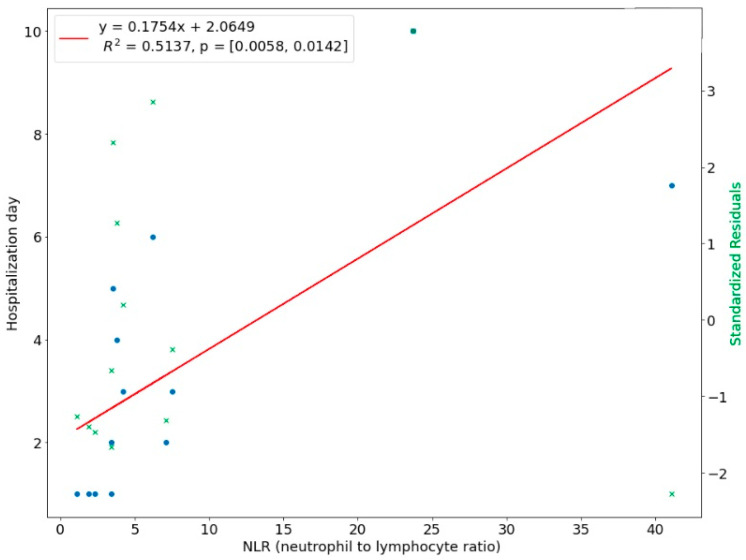
Correlation of NLR with length of hospital stay in upper face odontogenic infections.

**Figure 9 ijerph-20-04874-f009:**
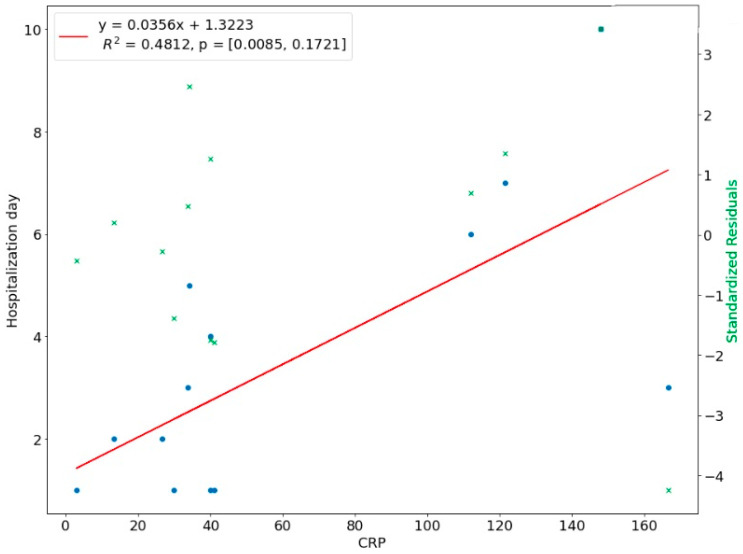
Correlation of CRP with length of hospital stay in upper face odontogenic infections.

**Figure 10 ijerph-20-04874-f010:**
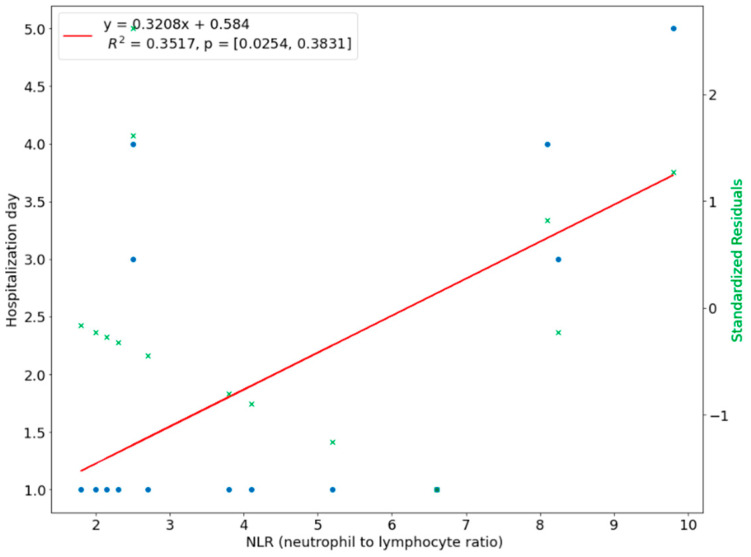
Correlation of NLR with length of hospital stay in lower face odontogenic infections.

**Figure 11 ijerph-20-04874-f011:**
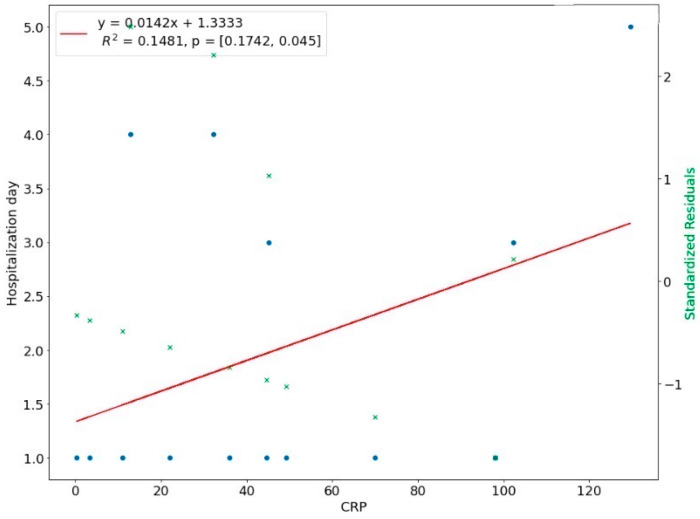
Correlation of CRP with length of hospital stay in lower face.

**Figure 12 ijerph-20-04874-f012:**
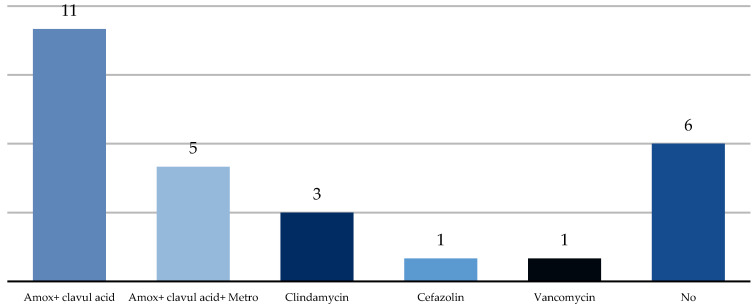
Antibiotics used for treatment.

**Table 1 ijerph-20-04874-t001:** Comparison of upper- and lower-face odontogenic infections.

Study Variables	Upper Face (n = 13)	Shapiro–Wilk Test	Lower Face (n = 14)	Shapiro–Wilk Test	*p*	Cohen’s d
Age	6.5 ± 5.39	non-normal (*p* ≤ 0.01)	10,14 ± 4.34	normal (*p* = 0.37)	0.076	0.74
Gender						
Male	7 (54%)		7 (50%)			
Female	6 (46%)		7 (50%)			
Length of hospitalization	3.54 ± 2.68	normal (*p* = 0.04)	2.0 ± 1.41	non-normal (*p* ≤ 0.01)	0.082	0.72
Source of infection						
Deciduous anterior	5 (38.5%)		0			
Deciduous posterior	5 (38.5%)		5 (36%)			
Permanent anterior	3 (23%)		0			
Permanent posterior	0		9 (64%)			
Symptoms						
Trismus	5 (38%)		10 (71%)			
Extraoral swelling	10 (77%)		11 (79%)			
Intraoral swelling	11 (85%)		14 (100%)			
Extraoral+ Intraoral swelling	10 (77%)		11 (79%)			
Fistula	6 (46%)		2 (14%)			
Fever > 37.5 °C	4 (31%)		3 (21%)			
Dysphagia	1 (8%)		6 (43%)			
Pain						
≥9 years old (VAS)	7.00 ± 0.81	normal (*p* = 0.999)	5.6 ± 0.92	normal (*p* = 0.2449)	0.051	1.62
<9 years old	10		4			
CRP [mg/L]	62.32 ± 52.24	normal (*p* ≤ 0.01)	46.9 ± 38.28	normal (*p* = 0.17)	0.41	0.34
NLR	8.4 ± 10.94	non-normal (*p* ≤ 0.01)	4.41 ± 2.61	normal (*p* = 0.02)	0.22	0.5
WBC [10^3^/uL]	12.44 ± 4.21	normal (*p* = 0.519)	9.53 ± 2.91	normal (*p* = 0.81)	0.54	0.8
D-Dimer [ng/mL]	1835.23 ± 3278.65	non-normal (*p* ≤ 0.01)	673.36 ± 513.75	non-normal (*p* ≤ 0.01)	0.22	0.5
Prealbumin [g/L]	0.125 ± 0.02	normal (*p* = 0.58)	0.14 ± 0.03	normal (*p* = 0.48)	0.11	0.59

**Table 2 ijerph-20-04874-t002:** Comparison of odontogenic infections of deciduous and permanent teeth.

Study Variables	Deciduous Teeth (n = 15)	Shapiro–Wilk Test	Permanent Teeth (n = 12)	Shapiro–Wilk Test	*p*	Cohen’s d
Age	4.4 ± 2.3	normal (*p* = 0.03)	13.75 ± 3.32	normal (*p* ≤ 0.01)	<0.01	3.27
Gender						
Male	8 (53%)		6 (50%)			
Female	7 (47%)		6 (50%)			
Length of hospitalization	2.2 ± 1.6	normal (*p* ≤ 0.01)	3.42 ± 2.72	normal (*p* = 0.02)	0.18	0.55
Source of infection						
Maxilla	10(73%)		3 (25%)			
Mandible	5 (27%)		9 ( 75%)			
Symptoms						
Trismus	3 (20%)		12 (100%)			
Extraoral swelling	9 (60%)		12 (100%)			
Intraoral swelling	13 (87%)		12 (100%)			
Extraoral+ Intraoral swelling	9 (60%)		12 (100%)			
Fistula	3 (20%)		5 (42%)			
Fever >37.5 °C	2 (13%)		5 (42%)			
Dysphagia	0		7 ( 58%)			
Pain						
≥9 years old (VAS)	5.0 ± 0		6 ± 1.08			0.55
<9 years old	14		0	normal (*p* = 0.6)		
CRP [mg/L]	38.09 ± 25.48	normal (*p* = 0.03)	74.62 ± 56.98	normal (*p* = 0.22)	0.04	0.83
NLR	3.38 ± 1.54	normal (*p* = 0.12)	10.03 ± 10.91	non-normal (*p* ≤ 0.01)	0.03	0.85
WBC [10^3^/uL]	10.68 ± 3.1	normal (*p* = 0.35)	11.24 ± 4.65	normal (*p* = 0.32)	0.72	0.14
D-Dimer [ng/mL]	639.93 ± 365.71	normal (*p* = 0.07)	1973.83 ± 3399.4	non-normal (*p* ≤ 0.01)	0.16	0.55
Prealbumin [g/L]	0.132 ± 0.025	normal (*p* = 0.07)	0.137 ± 0.031	normal (*p* = 0.5)	0.63	0.18

**Table 3 ijerph-20-04874-t003:** Gram-positive bacteria detected in post-extraction tooth socket.

G+	Bacterial Strain	Cultured Bacteria
Anaerobes (strict, facultative)		
	Actinomyces odontolitycus	1 (2.1%)
	Actinomyces oris	2 (4.2%)
	Actinomyces naeslundi	3 (6.4%)
	Corynebacterium accolens	1 (2.1%)
	Lactobacillus rhamnosus	1 (2.1%)
	Streptococcus anginosus	3 (6.4%)
	Streptococcus consellatus	2 (4.2%)
	Streptococcus gordoni	1 (2.1%)
	Streptococcus mitis	3 (6.4%)
	Streptococcus sanguinis	2 (4.2%)
	Staphylococcus aureus	4 (8.5%)
	Staphylococcus epidermidis	3 (6.4%)
	Staphylococcus capitis	1 (2.1%)
	Rothia dcentocariosa	1 (2.1%)
	Rothia mucilaginosa	1 (2.1%)
Sum		29 (62%)

**Table 4 ijerph-20-04874-t004:** Gram-negative bacteria detected in post-extraction tooth socket.

G-	Bacterial Strain	Cultured Bacteria
Anaerobes (strict, facultative)		
	Escherichia coli	2 (4.2%)
	Eikenella corrodens	1 (2.1%)
	Haemophilus parainfluenzae	3 (6.4%)
	Klebsiella pneumoniae	2 (4.2%)
Aerobes		
	Neisseria cinerea	1 (2.1%)
	Neisseria elongata	1 (2.1%)
	Neisseria macacae	2 (4.2%)
	Neisseria oralis	1 (2.1%)
	Neisseria subflava	5 (10.6%)
Sum		18 (38%)

## Data Availability

The data presented in this study are available on request from the corresponding author. The data are not publicly available due to privacy.

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
