# Peer review of "Analysis of the Clinical Status and Treatment of Facial Cellulitis of Odontogenic Origin in Pediatric Patients"

_ijerph, 2023, doi:10.3390/ijerph20064874_

Round 1
Reviewer 1 Report (Previous Reviewer 1)
Analysis of the Clinical Status and Treatment of Facial Cellulitis of Odontogenic Origin in Pediatric Patients
English needs review. Reviewed by authors
The article needs to be reviewed in some basic aspects. Some errors in English could easily be corrected in word or another text editor, and some double phrases too, like, for example:
"Table 2. Comparisons of odontogenic infections for primary and seondary teeth.
Lim et al. found that 24% of patients who reported to the their department had taken 238
The VAS scale was used in children over 9 185 years of age and in younger children we relied on an interview with parents The VAS 186 scale was used in children over 9 years of age, but in younger children we relied on an 187 interview with parents and their observation of the child at home. "
reviewed by authors
It is better always to use the same names when defining variables; permanent teeth instead of secondary teeth.
reviewed by authors
A critical review of methods
Sample
The number of patients is well documented in the methods section (n=20), but sample size calculations to meet significance are not shown. In the case of convenience sampling without a rigorous process of sample size calculation, there should be a paragraph in limitations explaining this.
– not reviewed by authors (09-02-2023), but it is not essential
Statistical analysis section
The statistical section is very simple, but referencing the statistical software used is important.
reviewed by authors (09-02-2023)
In the statistical analysis section, the authors wrote, "The Student's t-test was used to analyze the means and standard deviations in the specified groups." In this particular case, there should have been a sound reference to the normality of distributions because the sample size is small.
The authors wrote in the statistical analysis section, “The Student's t-test was used to analyze the means and standard deviations in the specified groups.” In this particular case, there should have been a sound reference to the normality of distributions because the sample size is small. – not reviewed by authors (09-02-2023)
Moreover, we suggest a more accurate way when writing about t-test:
"The Student's t-test was used to compare outcomes between the specified groups."
or
"The Student's t-test was used to compare dependent variables between the specified groups."
or
"The Student's t-test was used to assess the association of outcomes/dependent variables with upper-and lower-face odontogenic infections."
reviewed by authors (09-02-2023)
Results
In the results section, the effect sizes should have been reported; in this case, Cohen's d or Hedges g. Effect sizes are the best way to assess the magnitude of association between variables; p-values are very dependent on sample size.
reviewed by authors (09-02-2023)
Perhaps the paper could benefit from including significance calculations and effect sizes for the association of categorical variables (Tables 1 and 2). Standardized residuals can be calculated to identify significant cell differences (observed against expected values). Not reviewed by authors (09-02-2023)
The article's conclusions are an essential part of the article, should always be supported by the results, and must be revised. Not reviewed by authors (09-02-2023)
“Antibiotic therapy should be used only in 307 specific cases according to guidelines to reduce the risk of the appearance of antibiotic-308 resistant bacterial strains.” I believe this is true, but it cannot be a conclusion of this article.
The article could benefit from photos of facial cellulitis and swelling.
reviewed by authors (09-02-2023)
Turnitin was ok and did not detect plagiarism.
Author Response
Dear Editor and Reviewers
I would like to thank you and the reviewers for the careful evaluations of our manuscript entitled “Analysis of the Clinical Status and Treatment of Facial Cellulitis of Odontogenic Origin in Pediatric Patients.”(Submission. No. ijerph-2237757). The comments gave us an opportunity to further improve the quality of our manuscript. We have revised our manuscript in accordance with the comments provided. Responses to each comment are attached below in a point-to-point fashion, and all responses are highlighted in the manuscript. We hope that this revised version is now acceptable for publication in International Journal of Environmental Research and Public Health.
Sincerely yours,
Adrianna Słotwińska-Pawlaczyk, DDM
Prof. Dr Bogusława Orzechowska-Wylęgała
Katarzyna Latusek, DDM
Prof. Dr Anna Roszkowska

Reviewer 2 Report (New Reviewer)
This research topic is important for children's oral and general health.
However, the dental approach is considered to be a little insufficient.
Authors, Please supplement the contents below.
Introduction
1) Line 35-36 : which can cause the development of caries in perma- 35
nent teeth, while the primary location was in deciduous teeth
==> I don't understand this part.
Please check the previous research in dentistry and write clearly.
2) Line 38 : This is a very old data, so please replace it with a more recent one.
3) Line 48-49: Please add a reference.
4) I recommend you to put the Aim of the study at the end of the introduction.
Materials and Methods
1) The content of the methods is long. It is recommended to separate.
2) Considering the number of samples, the nonparametric statistic appears to be correct.
3) The number of samples seems insufficient to confirm Cohen's d.
Is there a reason you had to do this analysis? Please write at the limitation.
Results
1) The content of the results is long. It is recommended to separate.
2) Overall, tables and figures are not uniform, and variables and distinctions are unclear.
Please mark the table spacing, line type, and first letter in uppercase, and main variables (or parameters) separately.
p → p
Table 1 : Cohen’s d (0,72) → 0.72
Figure 1, 2 → n number notation
Figure 3,4 → What does the scale of the y-axis mean?
If you are going to compare 3 and 4, put them next to each other and make the scale the same.
Is there a relationship between odontogenic infection and alveoli infection?
It seems that the results came out of nowhere.
Please add the above to the introduction and research methods.
Discussion
1) What is a new discovery point in this study from previous studies?
The purpose of this study is ambiguous as it showed results almost similar to previous studies.
2) There is no information about alveoli infection described in the study results in the discussion.
3) Please add limitation.
Author Response
Dear Editor and Reviewers
I would like to thank you and the reviewers for the careful evaluations of our manuscript entitled “Analysis of the Clinical Status and Treatment of Facial Cellulitis of Odontogenic Origin in Pediatric Patients.”(Submission. No. ijerph-2237757). The comments gave us an opportunity to further improve the quality of our manuscript. We have revised our manuscript in accordance with the comments provided. Responses to each comment are attached below in a point-to-point fashion, and all responses are highlighted in the manuscript. We hope that this revised version is now acceptable for publication in International Journal of Environmental Research and Public Health.
.Sincerely yours,
Adrianna Słotwińska-Pawlaczyk, DDM
Prof. Dr Bogusława Orzechowska-Wylęgała
Katarzyna Latusek, DDM
Prof. Dr Anna Roszkowska

This manuscript is a resubmission of an earlier submission. The following is a list of the peer review reports and author responses from that submission.
Round 1
Reviewer 1 Report
Analysis of the Clinical Status and Treatment of Facial Cellulitis of Odontogenic Origin in Pediatric Patients
English needs review.
The article needs to be reviewed in some basic aspects. Some errors in English could easily be corrected in word or another text editor, and some double phrases too, like, for example:
"Table 2. Comparisons of odontogenic infections for primary and seondary teeth.
Lim et al. found that 24% of patients who reported to the their department had taken 238
The VAS scale was used in children over 9 185 years of age and in younger children we relied on an interview with parents The VAS 186 scale was used in children over 9 years of age, but in younger children we relied on an 187 interview with parents and their observation of the child at home. "
It is better always to use the same names when defining variables; permanent teeth instead of secondary teeth.
A critical review of methods
Sample
The number of patients is well documented in the methods section (n=20), but sample size calculations to meet significance are not shown. In the case of convenience sampling without a rigorous process of sample size calculation, there should be a paragraph in limitations explaining this.
Statistical analysis section
The statistical section is very simple, but a reference to the statistical software used is important.
The authors wrote in the statistical analysis section, "The Student's t-test was used to analyze the means and standard deviations in the specified groups." In this particular case, there should have been a sound reference to the normality of distributions because the sample size is small. Moreover, we suggest a more accurate way when writing about t-test:
"The Student's t-test was used to compare outcomes between the specified groups."
or
"The Student's t-test was used to compare dependent variables between the specified groups."
or
"The Student's t-test was used to assess the association of outcomes/dependent variables with upper-and lower-face odontogenic infections."
Results
In the results section, effect sizes should have been reported; in this case, Cohen's d or Hedges g. Effect sizes are the best way to assess the magnitude of association between variables; p-values are very dependent on sample size.
Perhaps the paper could benefit from including significance calculations and effect sizes for the association of categorical variables (Tables 1 and 2). Standardized residuals can be calculated to identify significant differences in cells (observed against expected values).
The article's conclusions are an essential part of the article, should always be supported by the results, and must be revised.
The article could benefit from the presence of photos of facial cellulitis and swelling.
Turnitin was ok and did not detect plagiarism.
Reviewer 2 Report
it seems good but your article can be improved by use of graphs instead of tables
Reviewer 3 Report
My main concern regarding this paper is the small sample size for a large age range. A sample of 20 is inadequate to make any conclusions.
Overall, there is no new finding in this manuscript .
Additional concerns
1. It is not clear if all patients were hospitalized
2. It is not clear why anterior permanent teeth were extracted and when the extractions were done.
3. The meaning of antilobium is not clear- what tool was used to measure? Ruler? Tape?
4. Post- operative Antibiotics were given to some patients, when were they started? Were they oral or IV? Why were they given to these specific individuals and not to others?
5. A separate category of those who had intra and extraoral swelling should be rerecorded.
6. Abbreviations such as NLR, CRP have been used throughout the document with no full form
Reviewer 4 Report
I applaud the authors on their effort whilst writing this manuscript, but I regret to inform you that in my opinion this paper is unsuitable for publication in its current for. The sample size is too small to derive any prevalence or incidence trend statistics, and so it the period of the study. A single year single institutional study does not add much to the literature. I would suggest accompanying these findings with global estimates and making rigorous comparisons may be more fruitful and clinically useful.
Rejections are always hard, but I do hope to review this paper again after a significant re-write up. I understand that English may not be the Authors first language, but I believe there are multiple free-editing services now available which could improve the readability of manuscripts as well.
I wish your team all the best.
Thanks